Improved estimates of age, growth and reproduction for the regionally endemic Galapagos sailfin grouper Mycteroperca olfax (Jenyns, 1840)

Usseglio Paolo 1 2 pusseglio@in-nova.org
Friedlander Alan M. 1 3
DeMartini Edward E. 4 5
Schuhbauer Anna 6 7
Schemmel Eva 1
Salinas de Léon Pelayo 6
1 Fisheries Ecology Research Lab, University of Hawaiʻi at Manoa , Honolulu, Hawaiʻi , USA
2 Fundacion In-Nova Castilla la Mancha , Toledo , España
3 Pristine Seas, National Geographic Society , Washington, DC , USA
4 NOAA Fisheries—Pacific Islands Fisheries Science Center, Fisheries Research and Monitoring Division, Life History Program , Honolulu, Hawaiʻi , USA
5 Hawaiʻi Institute of Marine Biology, University of Hawaiʻi, School of Earth Science and Technology , Kaneohe, Hawaiʻi , USA
6 Department of Marine Sciences, Charles Darwin Research Station , Galapagos Islands , Ecuador
7 University of British Columbia , Vancouver, British Columbia , Canada
Esteban María Ángeles
Electronic publication date: 2015 Sep 17
Publication date: 2015
Volume: 3
Electronic Location ID: e1270
Received 2015 Jun 12; Accepted 2015 Sep 3
Copyright year: 2015
License: This is an open access article, free of all copyright, made available under the Creative Commons Public Domain Dedication. This work may be freely reproduced, distributed, transmitted, modified, built upon, or otherwise used by anyone for any lawful purpose.
License URL: https://creativecommons.org/publicdomain/zero/1.0/

Keywords: Galapagos grouper, Life history, Growth, Age, Management, Maturity, Age and growth

Funding: Lindblad Expeditions-National Geographic Joint Fund for Conservation and Research Galapagos Conservation Trust The Mohammed Species Conservation Fund The Disney Worldwide Conservation Fund The Helmsley Charitable Trust Fisheries Ecology Research Lab University of Hawaiʻi This research was partly supported by grants from the Lindblad Expeditions-National Geographic Joint Fund for Conservation and Research, The Galapagos Conservation Trust, The Mohammed Species Conservation Fund, The Disney Worldwide Conservation Fund and The Helmsley Charitable Trust. Travel costs to Galapagos, as well as sample processing were self-funded by the lead author, sample processing materials and equipment were provided by the Fisheries Ecology Research Lab, University of Hawaiʻi. The funders had no role in study design, data collection and analysis, decision to publish, or preparation of the manuscript.

==============================
The Galapagos Sailfin grouper, Mycteroperca olfax, locally known as bacalao and listed as vulnerable by the IUCN, is culturally, economically, and ecologically important to the Galapagos archipelago and its people. It is regionally endemic to the Eastern Tropical Pacific, and, while an important fishery resource that has shown substantial declines in recent years, to date no effective management regulations are in place to ensure the sustainability of the Galapagos fishery for this species. Previous estimates of longevity and size at maturity for bacalao are inconsistent with estimates for congeners, which brings into question the accuracy of prior estimates. We set out to assess the age, growth, and reproductive biology of bacalao in order to provide more accurate life history information to inform more effective fisheries management for this species. The oldest fish in our sample was 21 years old, which is 2–3 times greater than previously reported estimates of longevity. Parameter estimates for the von Bertalanffy growth function (k = 0.11, L∞ = 110 cm TL, and to = − 1.7 years) show bacalao to grow much slower and attain substantially larger asymptotic maximum length than previous studies. Mean size at maturity (as female) was estimated at 65.3 cm TL, corresponding to a mean age of 6.5 years. We found that sex ratios were extremely female biased (0.009 M:1F), with a large majority of the individuals in our experimental catch being immature (79%). Our results show that bacalao grow slower, live longer, and mature at a much larger size and greater age than previously thought, with very few mature males in the population. These findings have important implications for the fishery of this valuable species and provide the impetus for a long-overdue species management plan to ensure its long-term sustainability.

Introduction

The Galapagos Sailfin grouper, Mycteroperca olfax, has high cultural, economic, and ecological importance to the people and the marine ecosystem of the Galapagos Archipelago. It is one of the most sought after species in the artisanal hand-line fishery and is prized in its dried form for a traditional dish called “fanesca” that is consumed during Easter (Nicolaides et al., 2002). With a trophic level of 4.2, bacalao is the top demersal predator in the Galapagos (Okey et al., 2004). The species is likely a protogynous hermaphrodite (Rodriguez, 1984; Coello & Grimm, 1993), and it is regionally endemic to the Eastern Tropical Pacific (ETP), where it is commonly found throughout the Galapagos, and to lesser extent at Cocos Island, Costa Rica, and Malpelo Island off the west coast of Colombia (Grove & Lavenberg, 1997), although no fishery exists for this species at these locations.

The artisanal hand-line fishery that targets M. olfax dates back to the late 1920s, when Norwegians introduced the butterfly method of salting and drying fish that gives bacalao (cod in Spanish) its name (Reck, 1983). Over the years bacalao has been one of the most sought after species, representing almost 100% of the finfish landings in the 1940s (Reck, 1983), to 89% of the total finfish catch in the 1970s, and only 17% in recent years (Reck, 1983; Nicolaides et al., 2002; Schiller et al., 2014). A very limited geographical range, and clear evidence of fisheries declines, have led the International Union for the Conservation of Nature (IUCN) to list bacalao as Vulnerable (VU) (Bertoncini et al., 2008).

Fishing regulations in the Galapagos Marine Reserve include a zonation scheme, with no-take areas (Heylings, Bensted-Smith & Altamirano, 2002), a licensing system, gear restrictions (e.g., spearfishing and long-lining), a ban on industrial fishing vessels, and a ban on capture and marketing of sharks (Castrejón et al., 2014). However, these regulations are poorly enforced and not well adhered to by many in the fishing community, which has resulted in overfishing of a number of prized species (Bustamante et al., 2000; Ruttenberg, 2001; Edgar et al., 2010). While bacalao has historically been the most important finfish fishery in the Galapagos, and despite suggestions of overfishing (Schiller et al., 2014), to date there are no management regulations specifically aimed at this fishery (e.g., total allowable catch, size limits, fishing seasons).

Fisheries are often managed to lessen the consequences of uncontrolled fishing, which can lead to the collapse of a fishery, economic inefficiency, loss of employment, habitat degradation or decreases in the abundance of rare species (Jennings, Kaiser & Reynolds, 2009). Fishery models use basic life history information (e.g., age, growth, reproduction) as inputs to determine how species respond to fishing. Models such as virtual population analysis are used to calculate mortality rates of age-based cohorts, making it critical to accurately assign the correct age to an individual of a given size. However, in the case of bacalao, previous studies have provided conflicting results, and many of the life history characteristics for this species remain uncertain. For example, the sexual pattern for this species has not been resolved, while previous studies claim the species to be hermaphroditic (Rodriguez, 1984; Coello & Grimm, 1993), these claims are not supported by histological analysis. In terms of longevity, otolith-derived age estimates range from 7 to 11 years (Rodriguez, 1984; Gagern, 2009), which seems low for a grouper that attains a maximum length of 120 cm TL. Similarly, various studies have provided dissimilar estimates of size at sexual maturity that range from 47 to 67 cm TL (Rodriguez, 1984; Coello, 1989; Heemstra & Randall, 1993). These contrasting results can influence fishery models. For example, underestimation of longevity leads to overestimation of growth and mortality rates (Mills & Beamish, 1980), which can in turn affect the outcome of population models used to formulate management for fish stocks (Tyler, Beamish & McFarIane, 1989; Reeves, 2003).

Because of the uncertainty in important life history parameters, our goal was to determine the longevity, growth rate, and size at maturity of bacalao in the Galapagos Islands in order to provide more accurate information for better management of this species.

Materials & Methods

Study site

The Galapagos Archipelago is located 1,000 km off the coast of Ecuador, and comprises thirteen islands and over 100 islets (Snell, Stone & Snell, 1996). The Galapagos Marine Reserve (GMR) encompasses approximately 133,000 km2 and was the first marine reserve established in Ecuador in 1998, being recognized in 2001 as a UNESCO World Heritage Site (Heylings, Bensted-Smith & Altamirano, 2002). While there is a ban on industrial fishing vessels in the GMR, artisanal fishing is allowed in fishing areas delimited by the GMR zonation scheme (Castrejón et al., 2014). Despite this status, artisanal fishing occurs throughout much of the archipelago.

Sample collection

Landings of bacalao were assessed at the fishing port of Pelican Bay on the island of Santa Cruz, which is the major landing port on the island. Measurements for every bacalao landed were taken, and opportunistic sampling of otoliths was performed. In addition, we conducted experimental fishing trips by accompanying local fishers. Fishers would select their regular spots around the islands of Santa Cruz, Santiago, Isabela, and Wolf (Fig. 1), and conduct fishing the same way as a typical commercial trip. All fish were processed (e.g,: removal of otoliths and gonads and measurements) before the fisher took them to market. To ensure that our sampling represented the landings from the fishing community, fishers employed the traditional hook and line method, “empate,” which is homogeneously used among all fishers. Examination of results from previous studies shows that this fishing method captures bacalao from 19 to 100 cm TL, suggesting that gear selectivity did not affect size composition of the landings, and that the catch is representative of the population. Otoliths were collected in June 2011, February 2012, and from September 2012 to April 2013, whereas tissue samples for histological analysis were collected from October 2012 to February 2013, in order to coincide with the identified spawning period for this species (Coello & Grimm, 1993). All samples were collected under Galapagos National Park field permits PC-19-11, PC-24-13, and PC-25-14, and animal use approval by the Animal Care & Use Committee, University of Hawaiʻi, protocol number 11-1284.

Figure 1 Study area.

Study area of the Galapagos archipelago, red dots represent sampling sites; inset map shows the location of the Galapagos archipelago.

Sample processing

Total (TL) and standard length (SL) of bacalao were measured to the nearest cm, and total weight was taken to the nearest gram. Sagittal otoliths were extracted from each fish, washed in water and cleaned in 95% ethanol prior to dry-storage. The right sagittal otolith was weighed to the nearest 0.1 g and its maximum length and width recorded to the nearest 0.1 mm. Otoliths were then fastened to a 2 × 2 cm square of plywood using acrylic glue, and a single transverse section, through the primordium, ∼300 µm thick was cut at low speed using a Buehler™ IsoMet saw with two parallel blades separated by a shim. Sections were then mounted on glass slides using clear adhesive (Crystalbond™), ground for a few seconds with sand paper 400 grit, polished with 15 µm lapping film, and viewed immersed in water against a dark background at low power on a dissecting microscope using two reflected light sources angled at 45°. The number of clear bands observed in the transverse sections was counted, as well as the appearance of the outer margin (clear vs. opaque).

Gonads were removed from the fish and weighed to the nearest 0.1 g. A 0.5 cm transverse section from the middle of the gonad was cut and placed inside a tissue-embedding cassette before being fixed in 10% buffered formalin for 7 days, after which the samples were washed overnight in water to remove formaldehyde crystals, then worked up to 70% ethanol through a series of dilutions. Tissue samples were embedded in plastic resin (JB-4™) following previously established protocols (Sullivan-Brown, Bisher & Burdine, 2011). Embedded samples were sectioned into 3 µm slices using an automated microtome with a glass blade. Following sectioning, samples were mounted on glass slides, stained with toluidine blue and viewed under a dissecting microscope using transmitted light at 100× magnification (Olympus BX41, BX41TF). Samples were assigned to reproductive stages following key characteristics in the reproductive cycle previously proposed by Brown-Peterson et al. (2011) (Table 1).

Table 1 Reproductive phases.

Reproductive phases, histological features, and maturity rating used to assess the maturity of female bacalao. Phases follow key milestones in the reproductive cycle (modified from Brown-Peterson et al., 2011).

Phase	Histological features	Maturation	
Immature (never spawned)	Small ovaries, often clear, blood vessels indistinct. Only oogonia and primary growth oocytes present. No atresia or muscle bundles. Thin ovarian wall and little space between oocytes.	Immature	
Developing (ovaries developing but not yet ready to spawn)	Enlarging ovaries, blood vessels becoming more distinct. Primary growth, cortical alveolar, vitellogenic stages 1 or 2. No evidence of vitellogenic stage 3 or postovulatory follicles.	Mature	
Spawning capable	Large ovaries, blood vessels prominent, Vitellogenic stage 3 oocytes and postovulatory follicles present. Early stage maturation oocytes might be present. Atresia or early vitellogenic oocytes might be present.	Mature	
Regressing	Atresia at any stage, postovulatory follicles present. Blood vessels prominent. Some cortical alveolar and or vitellogenic oocytes stages 1 or 2 present.	Mature	
Regenerating	Small ovaries, blood vessels reduced but present. Only oogonia and primary growth oocytes present. Muscle bundles, enlarged blood vessels, thick ovarian wall, atresia, old degenerating postovulatory follicles, may be present.	Mature	

Data analysis

Age and growth

Ageing consisted of assigning each fish a count of annual growth increments that comprised a ring of alternating clear and opaque sections. These were counted with no reference to time of collection or fish length. Each otolith was read twice by a single observer, and the coefficient of variation (CV) was calculated (Campana, Annand & McMillan, 1995). For otoliths where the CV was greater than 10%, the otolith was read a third time. If no consensus was reached among counts, or if the otolith appeared irregular, or had poorly defined growth increments, the otolith was discarded from all further analyses. Validation of the first increment width was conducted by performing daily increment counts.

Growth was modeled using the von Bertalanffy growth function (Bertalanffy, 1938): Lt=L∞1−e−kt−to+ε

where Lt is the predicted mean length at age t, L∞ is the asymptotic mean length, k is the Brody growth coefficient, t0 is the theoretical age at which length is 0, and ε denotes the belief that residuals would be distributed normally about the expected growth line (Haddon, 2010). Estimated ages were adjusted to decimal age by assigning a birth date for all fish set at October 1st, which was the beginning of the spawning season. Starting parameters for the model were determined using a Ford-Walford plot, model parameters were estimated using nonlinear (weighted) least-squares by means of the NLS function in R. Confidence intervals for the resulting model parameter estimates were calculated via bootstrapping with 1,000 iterations. These analyses were conducted using the R package FSA (Ogle, 2015).

The length–weight relationship for bacalao was obtained by fitting the power function W = aLb to weight and length data where: W is the total wet weight, L total length, and a and b are empirically derived constants. Length and weight data were log (natural) transformed, and transformation bias of the scale parameter was corrected using a = exp(b)∗exp(σ2/2) as a correction factor, where b is the intercept parameter of the model, and σ2 is the estimated residual variance of the regression model (Hayes, Brodziak & O’Gorman, 1995).

Reproduction

The adult sex ratio was calculated using all sexually mature females and males. A chi square test was used to assess whether sex ratios differed significantly from 1:1. Mean size at sexual maturity (L50) was calculated by fitting a logistic model to the proportion of mature fish binned in 5 cm size classes. The logistic model follows the formula: log(p/1 − p) = a + βTL, where p is the probability of being mature, TL is total length, and a and β are fitting constants. Proportion of mature fish were fitted by iteratively reweighted least squares (IWLS) using the GLM function in R (R Development Core Team, 2013). Confidence intervals for the predicted model parameters were estimated via bootstrapping with 1,000 iterations. Mean age at first maturity was estimated using the same method. Additionally, an estimate of L50 was obtained from the empirical relationship between L50 and L∞ derived using the equation log L50 = 0.9469∗log L∞ − 0.1162 (Froese, 2000). Gonadosomatic index (GSI) was calculated as GSI = (GW/SW)∗100%, where GW is the wet gonad weight, and SW is the somatic weight, or gonad free body weight, this metric was calculated only for mature fish.

All analyses were done using the statistical software R (R Development Core Team, 2013). Data manipulation was performed using the reshape package (Wickham, 2007), and graphing was conducted with the package ggplot2 (Wickham, 2009).

Results

Sampling of landings at the fishing port, as well as samples taken from fishers, resulted in a total of 297 bacalao, with a mean TL of 49.8 cm (±11.1 sd), and a range from 18 to 100 cm TL (Fig. 2). The length (TL cm) to weight (g) relationship, calculated from these samples, resulted in estimates for a = 5.47e−6 and b = 3.158, the coefficient of determination of the log transformed data (natural log) was r2 = 0.94.

Figure 2 Size class distribution of fish used to assess growth.

Size class distribution of otolith sub-sample fish (n = 141). Black dashed line represents mean TL (50.4 cm ± 12.8 sd).

Unpublished reports from the Charles Darwin Foundation that monitored bacalao during the peak fishing times for the 2011 fishing season (January and August–November) resulted in a total of 277 fish landed at the port of Pelican Bay. Our sampling of 297 bacalao is therefore representative of the fishery landings as a whole.

Age and growth

From the collected otoliths, only 198 were readable and used for growth analysis. Sectioned sagittal otoliths, when viewed under a dissecting microscope with reflected light, showed alternating clear and opaque bands (Fig. 3). Validation of the first annuli, by counting daily rings, was limited by the number of readable rings in the sample analyzed. While it was not possible to read the section nearest the core, an estimate of 280 days was produced by interpolating unreadable spaces with the mean width of the observable bands. The first annuli was estimated to occur at ∼511 µm from the core.

Figure 3 Photomicrograph of otolith cross section.

From a 49 cm, 5 year old bacalao, as viewed through a dissecting scope against a dark background with reflected light from two sources placed at opposing sides at a 45° angle. Red dots represent yearly rings.

Growth modeling and length at age

Age estimates from otoliths ranged from 1 to 21 years. The estimated von Bertalanffy growth model parameters were k = 0.11 (95% CI [0.09–0.15]), L∞ = 110 cm (TL) (95% CI [99–125 cm TL]), and to = − 1.7 years (95% CI [−2.3 to −1.2]) (Fig. 4).

Figure 4 von Bertalanffy growth function.

The von Bertalanffy growth function fit to size at age data for bacalao in the Galapagos (n = 198).

Reproduction

Sampling of bacalao during the reproductive season (October–February) resulted in 116 fish with a size range between 34 and 81 cm TL and a mean TL of 51.4 cm (±10.4 sd) (Fig. 5). The sex ratio was significantly female skewed (χ2 = 112, p < 0.01), with 0.009 males per female.

Figure 5 Size composition of bacalao sampled for gonads.

Size composition of bacalao sampled for gonads, by sex (n = 116). Black dashed line represents mean TL (51.4 cm ± 10.4 sd).

Immature fish represented 79% of the sampled individuals (n = 92) (Fig. 6A) and had a mean TL of 48.9 (±8.3 sd) cm. Mature bacalao, including fish that were developing, spawning capable or that were regressing and whose ovaries contained regenerating oocyte stages (Figs. 6B and 6C), represented the remaining 21% of the samples (n = 24). These fish had a mean TL of 60.6 (±11.3 sd) cm. There was only one male in our samples (Fig. 6D), with a size of 81 cm TL. Due to the high occurrence of immature females in our sample collection, we were not able to obtain enough mature samples to accurately determine spawning peaks or spawning times. GSI reached the highest value for the month of December, when seawater temperatures started warming up, however we did not have samples for the month of January (Fig. 7).

Figure 6 Photomicrograph of toluidine blue stained gonad cross sections of Mycteroperca olfax.

(A) Immature 46 cm TL female exhibiting primary growth oocytes (PG), (B) spawning capable 70 cm TL female with oocytes in vitellogenic stage 3 (Vt3) and atresia (At), (C) spawning capable 84 cm TL female with oocytes in vitellogenic stage 3 (Vt3) and hydrated (Hy), (D) male 98 cm TL with spermatocrypts (Sc), and spermatozoa (S). Scale bars 100 µm.

Figure 7 Developmental stages by month.

Proportion of developmental stages of Mycteroperca olfax sampled by month, (A) represents the n for each month. (B) Box and whisker plot of gonadosomatic index (GSI) for fish sampled, line represents the median, box are the lower and upper quartiles (25% and 75%), points represent the outliers, and numbers above represent the number of mature fish per month. (C) Mean water temperature at the island of Santa Cruz.

Size at which 50% of the population reached sexual maturity (L50) was estimated from the logistic regression model as 65.3 cm TL (95% CI [61.3–74.9]), while age at maturity was 6.5 years (95% CI [5.7–7.8], Fig. 8). The empirical equation proposed by Froese (2000) estimated size at maturity for females as 65.6 cm TL (49.5–86.9 se). Since we only sampled one male, it was not possible to estimate size or age at sex change.

Figure 8 Ogive for size and age of sexual maturity.

Maturity ogives for female bacalao (n = 116), (A) size (TL) at which 50% of the population matures (L50 = 65.3 cm TL, 95% CI [61.3–74.9]), (B) age at which 50% of the population matures (6.5 years, 95% CI [5.7–7.8]). Blue dashed lines represent size and age at which 50% of the population matures; red line is the resulting logistic model, black squares are proportions of categorized values.

Discussion

Our estimates of maximum age of bacalao (21 years) were two to three times higher than those previously reported. The large proportion of immature individuals in our samples, as well as the low number of larger individuals, and highly biased sex ratio suggests that the resource has undergone, and is probably still experiencing, severe overfishing. We have provided more accurate estimates of size-at-age, growth, and size and age at sex change of bacalao, and these estimates should be urgently incorporated into management plans for this species.

Age and growth

Previous age estimates of bacalao ranged between 7 and 11 years (Rodriguez, 1984; Gagern, 2009). Longevity for other mycteropercids such as Mycteroperca bonaci (Max TL 150 cm) is 34 years, while M. macrolepis (Max TL 145 cm) and M. phenax (Max TL 107 cm) reach 22 and 21 years, respectively (Froese & Pauly, 2015). Our maximum recorded age of 21 years is closer to what would be expected of a mycteropercid grouper, and while our biggest fish was only 100 cm TL, bacalao is reported to reach 120 cm TL (Walford, 1937), suggesting that it is likely that longevity for this species is closer to 30 years. Differences in age between our study and previous works very likely stem from difficulties in reading bacalao otoliths. Rodriguez (1984) reported that 90% of the otoliths collected were considered unreadable due to the presence of a large number of false rings, or the lack of rings that he attributed to demineralization. Similarly, Gagern (2009) reported finding a large number of rings that he presumed were formed on a monthly basis. It was clear in the present study that bacalao otoliths are not necessarily easy to interpret, but we found that cross-sections can be read more easily using a dissecting microscope with reflected light against a dark background than with transmitted light and a compound microscope. Both Rodriguez (1984) and Gagern (2009) employed transmitted light and a compound microscope for their readings.

Estimation of age and growth in marine teleosts is useful for a variety of purposes such as estimating mortality (Pauly, 1980), predicting responses to exploitation (Jennings, Reynolds & Mills, 1998; Jennings, Kaiser & Reynolds, 2009), and developing fisheries management arrangements (Frisk, Miller & Dulvy, 2005). However, inaccurate estimates of these key parameters can result in models that do not accurately represent exploitation of the species. For instance, in the case of the orange roughy (Hoplostethus atlanticus), initial estimates of longevity were 24 years, while later validated estimates were nearly 100 years (Andrews, Tracey & Dunn, 2009), resulting in dramatically different estimates of natural mortality and exploitation rates (Tracey & Horn, 1999). It is important to note, however, that our estimates of growth parameters are based on sampling that is limited in smaller (<20 cm TL) and larger (>65 cm TL) individuals, so caution is needed in the use of these data and their application. Additional sampling in the future focusing on these size gaps may generate more robust estimates. Nonetheless, large individuals in particular are notoriously rare due to the overfishing experienced by the species.

Reproduction

While this study aimed to resolve some of the uncertainties regarding reproduction of bacalao, logistic constraints as well as the inherent difficulty in securing enough samples of reproductive age individuals precludes final conclusions in many aspects. We were not able to collect any transitional individuals that would resolve the sexual pattern for the species. While there have been suggestions that bacalao is protogynous (Rodriguez, 1984; Coello & Grimm, 1993), these studies relied on gross morphology of the gonads, rather than histology, which is known to produce inaccurate estimates of reproductive size (West, 1990) and even of sexual identity in protogynous groupers (DeMartini, Everson & Nichols, 2011). Because detailed microscopic examination of gonads is necessary to determine the sexual pattern in most fishes (Sadovy & Shapiro, 1987), these findings cannot be considered conclusive.

Most groupers are monandric protogynous hermaphrodites, maturing as females and then switching to males (Smith, 1959; Thompson & Munro, 1974; Collins et al., 1987; Sadovy, Figuerola & Roman, 1992), however there are exceptions such as the Nassau grouper (Sadovy & Colin, 1995; Chan & Sadovy, 2002). While the pattern for bacalao is still unresolved, it is likely that the species is protogynous and follows monandry. These assumptions are supported by the observable patterns of a lack of males at smaller sizes, and the severely skewed sex ratios observed by us and other authors in the past. However, it is important to stress that until histological evidence is provided, the sexual pattern should remain unresolved. The conundrum lies in that the observed pattern of very low abundance of larger size classes will only be exacerbated by continued heavy fishing pressure.

Reproductive periodicity in bacalao has been suggested to peak in the months of October–January (Coello & Grimm, 1993), and December and April (Rodriguez, 1984). While logistical constraints restricted our sampling to the period of October–February, the GSI attained the highest values in the month of December. Additionally, the first recorded evidence of a spawning aggregation for bacalao was in the month of November (Salinas-de-León, Rastoin & Acuña-Marrero, 2015). Although there is not clear consensus on reproductive periodicity, sufficient information exists to establish a closed season from October to April, which would protect bacalao during much of the spawning season.

Information on size at maturity for bacalao has been highly contradictory, with Coello (1989) reporting size at maturity at 47.5 cm TL, and Rodriguez (1984) reporting it at 63 cm TL. However, both of these studies used macroscopic examination, which is known to produce inaccurate estimates as discussed above. Fishbase (Froese & Pauly, 2015), reports 67 cm, citing Heemstra & Randall (1993) as the source, we however, did not find any reference to size at maturity for bacalao in the latter paper. Our sampling was focused during the peaks of reproduction for the species to ensure that the maximum amount of mature individuals were collected, and found a size at maturity larger than those previously reported (65 cm TL). Our estimates are consistent with those published by Rodriguez (1984), and the empirical estimates derived from the equation of Froese (65 cm) (2000), although it is recognized that the predictive model of Froese (2000) has a high level of uncertainty.

Bacalao has been suggested to have experienced severe overfishing as early as the 1950s (Reck, 1983), which could have resulted in fishing induced changes in life history (Conover & Munch, 2002; Baskett et al., 2005; Enberg et al., 2012; Sharpe, Wandera & Chapman, 2012). Evidence of overfishing includes a decrease in mean landed size since the 1970s (Reck, 1983; Nicolaides et al., 2002), a marked increase in the proportion of immature fish in landings from 35% in 1993 (Coello & Grimm, 1993) to 73% in our study, and an overall decline in the catch from 89% of the total finfish catch in the 1970’s to only 17% in recent years (Reck, 1983; Nicolaides et al., 2002; Schiller et al., 2014). However, the decline in contribution of bacalao to the overall catch is partially a result of diversification of the fisheries and a shift to pelagic species such as tuna and wahoo (Schiller et al., 2014), that reflects the preference for these species by the local tourist industry and export market, which may mask the effect of overfishing. While this market preference can explain some of the declines, the effect of artisanal fishing in the declines of bacalao has already been demonstrated for the Galapagos (Ruttenberg, 2001). Lacking reproductive biology data from earlier periods (before the 1950s) will make it impossible to resolve the reproductive patter for a non-exploited population.

Fishery effects on bacalao quite likely also include impacts on the adult sex ratio and size-at-sex change. The highly skewed sex ratio we observed (0.015 M:F) is consistent with those previously reported (0.021: Coello & Grimm, 1993; 0.017: Reck, 1983), suggesting that the adult sex ratio had likely been altered by fishing down of the larger bacalao prior to the 1980s, especially by targeting spawning aggregations (Salinas-de-León, Rastoin & Acuña-Marrero, 2015). Although there has not been a decline in catch rates in the bacalao fishery over time (Nicolaides et al., 2002), this might represent expansion of the fishery to new unfished areas. In the 1980s fishers reported that larger fish were caught mostly on “far” offshore banks such as Banco San Luis, which is located only 30 km from the main fishing port on Santa Cruz Island (Reck, 1983). In contrast, the largest fish in our study, as well as two additional males sampled for otoliths were caught at Wolf Island, located in the far north bio-region of the GMR, nearly 300 km from Santa Cruz Island.

Size-selective fishing mortality typically results in the differential loss of larger and older males in protogynous groupers (Sadovy, 1996). Major implications of severely skewed sex ratios include reduction in the probability that females will survive to sex change (Armsworth, 2001), loss of productivity due to sperm limitation (Bannerot et al., 1987; Koenig et al., 1996), and ultimately reproductive failure as males become too rare to effectively mate with females (Allee effect) (Bannerot et al., 1987; Huntsman & Schaaf, 1994).

Conclusions

Protogynous species have been suggested to be at particular risk of overexploitation, even when fishing mortality rates are low (Huntsman & Schaaf, 1994; Alonzo & Mangel, 2004; Heppell et al., 2006). This is especially relevant in an area such as the Galapagos where there are no management regulations specific to bacalao (e.g., allowable catch, size limits, fishing seasons). While the sexual pattern of bacalao remains unresolved, it is likely that the species is hermaphroditic, therefore, and heeding the precautionary approach, conservative management approaches for the species should include a mix of control over catch and fishing effort such as slot limits (Heppell et al., 2005), seasonal closures during reproductive season, and realized no-take spatial closures (Beets & Friedlander, 1999; Sadovy, 2001; Sadovy & Domeier, 2005; Heppell et al., 2006; De Mitcheson et al., 2008). Currently, the only protection for bacalao occurs in no-fishing zones which are yet to show positive evidence of protection (Nicolaides et al., 2002). The results from this paper provide needed inputs for fisheries models in order to determine adequate levels of catch and fishing effort that would ensure the long term sustainability of the bacalao fishery and reduce current levels of discards and bycatch associated with it (Zimmerhackel et al., 2015). Creating management regulations, however, is often easier than implementing them. This is especially true in developing countries (McClanahan, Maina & Davies, 2005; King, 2013), where using collaborative approaches is necessary to ensure compliance to management regulations (Yochum, Starr & Wendt, 2011; Usseglio, Schuhbauer & Friedlander, 2014). By working with the local fishing community to develop more accurate estimates of age, growth, and reproduction of bacalao, our results are more credible to them, and therefore more likely to be accepted in any future management decisions.

Supplemental Information

Supplemental Information 1 Age of bacalao

Click here for additional data file.

Supplemental Information 2 Reproduction dataset

Click here for additional data file.

Supplemental Information 3 Temperature dataset

Click here for additional data file.

We thank Jorge Baque, and staff at the CDF who contributed to fieldwork and sample processing. Dr. Thomas Helser and Delsa Anderl (Age and Growth Program, Research Ecology and Fisheries Management Division, Alaska Fisheries Science Center, NOAA Fisheries) for help with validation of age estimates, and Ryan S. Nichols and Matt Craig for help and guidance in otolith preparation and reading. We also thank the artisanal fishers of Galapagos for their help, and our special thanks go to Don Gabino and Marco Gabino for their support and enthusiasm. Lastly, sample processing and analysis would not have been possible without the logistic support of the Fisheries Ecology Research Lab at the University of Hawaiʻi.

Additional Information and Declarations

Competing Interests

Author Contributions

Animal Ethics

Field Study Permissions

The authors declare there are no competing interests. Alan M. Friedlander is an employee of Pristine Seas, National Geographic Society.

Paolo Usseglio conceived and designed the experiments, performed the experiments, analyzed the data, contributed reagents/materials/analysis tools, wrote the paper, prepared figures and/or tables, reviewed drafts of the paper.

Alan M. Friedlander conceived and designed the experiments, performed the experiments, analyzed the data, contributed reagents/materials/analysis tools, wrote the paper, reviewed drafts of the paper.

Edward E. DeMartini wrote the paper, reviewed drafts of the paper.

Anna Schuhbauer conceived and designed the experiments, performed the experiments, contributed reagents/materials/analysis tools, reviewed drafts of the paper.

Eva Schemmel analyzed the data, contributed reagents/materials/analysis tools, wrote the paper, reviewed drafts of the paper.

Pelayo Salinas de Léon performed the experiments, contributed reagents/materials/analysis tools, reviewed drafts of the paper.

The following information was supplied relating to ethical approvals (i.e., approving body and any reference numbers):

Protocol 11-1284 from the Animal Welfare and Use Program, University of Hawaiʻi.

The following information was supplied relating to field study approvals (i.e., approving body and any reference numbers):

This project was conducted under permits: PC-19-11, PC-24-13, and PC-25-14 from the Galapagos National Park.

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
