# Peer review of "Improved estimates of age, growth and reproduction for the regionally endemic Galapagos sailfin grouper Mycteroperca olfax (Jenyns, 1840)"

_PeerJ, doi:10.7717/peerj.1270_

## Round 0.1 · original submission · Major Revisions

Please consider the suggestions of the reviewers, in particular Reviewer 1, in your revised version.

Reviewer 1 ·

Basic reporting

No comments

Experimental design

No comments

Validity of the findings

I have no serious problems with the findings from the authors' work, I do have some problems in the designation of sexual pattern for this species, as this has not been fully resolved in this or any prior study (see below in Comments for the Author). This impacts what management recommendations can be made with regards to specific management measures, such as size limits.

I am also surprised not to find any information in the results of reproductive periodicity. While it is clear that no samples were taken in July and August, it appears that there should be sufficient information for females to show when the various developmental stages appeared within months and over the sampling period, and in relation to water temperature. If this information is not available from the sampled fish, the report can't be considered comprehensive.

Additional comments

I have a basic problem with the findings in that the authors make a broad assumption that this species is indeed a protogynous hermaphrodite. Both of the references cited (Rodriguez and Coello and Grimm) proclaim this species to be hermaphroditic, yet showed no clear histological proof that the sexual pattern is hermaphroditism nor did the authors. For this reason, I urge extreme caution in using this definition of the species’ sexual pattern in the manuscript and prefer a qualifier. While I would not necessarily disagree that the absence of males in the smaller size classes suggests that the species likely undergoes sexual transition, I would urge the authors to describe the sexual pattern as ‘unresolved’ and not confine management recommendations to only hermaphrodites. Given that the population appears to be highly overfished, particularly those fish in the upper size classes, which are presumably male, the sexual pattern may never be resolved, at least not for Galapagos stocks. I also urge the authors to state that if the species is actually a protogynous species, it’s male development pathway appears to follow monandry, which has not been mentioned in any of the references or within the current manuscript.

Even if the male development pathway is monandric, I find it curious that there were no males in the smaller size classes, given that the first size at sexual maturation appears to occur at or below 40 cm TL and there appear to be a number of samples between 40 cm and the upper size class sampled. Based on the evidence provided, sexual transition, should it occur, only occurs in larger, older females, which is not typically the case in sex changing groupers. In other words, I would expect a smattering of males in lower size classes, even with the moderately small sample size. Could this absence partly be a function of the logistical constraints on performing histology on the entire sample, since nearly 1/3rd of the individuals were not examined?

Why is there no graphic information on developmental stages by month within the sample period? The manuscript clearly shows that ripe females were sampled, but when? Graphic depiction of developmental stages by months helps the reader understand when this species is spawning, when it is resting, etc. I would expect this to be a centerpiece of the manuscript, especially given that the authors use the word 'comprehensive' in the title. Sample numbers by month should be included in any such graph.

Finally, I would suggest the authors also consider mentioning slot limits, even though size limits are perhaps the most difficult of the possible management measures in a developing country setting. The protection from fishing of larger individuals may permit some males being maintained within the population. Specific mention should also be made of seasonal closures during reproductive periods, given that the spawning times appear to be fairly well known.

Reviewer 2 ·

Basic reporting

It is not possible to assess whether the submission adheres to all PeerJ policies (e.g., corresponding author’s role and responsibilities). The article is in English and is grammatically sound, clear and, in general, unambiguous. In describing the state of knowledge regarding the age, growth, and reproductive characteristics of Mycteroperca olfax in lines 84–100, however, it appears that the estimates of longevity and size at sexual maturity may relate primarily to species other than M. olfax. It would be useful in this paragraph to identify more clearly what is known about the biological characteristics of the population of M. olfax in the waters of the Galapogas Islands, itself, and whether the basis for age estimation in those earlier studies was scales or otoliths. To conform with the structure of the paper recommended in the guide to authors, the heading ‘Methods’ at line 105 should be modified to ‘Materials & Methods’, and the heading ‘Acknowledgments’ at line 327 should be written as ‘Acknowledgements’. Contrary to the advice in the guide to authors, the latter section currently acknowledges funders and should be modified accordingly.
It would be useful in Fig. 1 to provide tick marks on the border to identify the latitudinal and longitudinal range of the study location. In Fig. 2 (and Fig. 6), the title of the y-axis should read ‘Frequency’, and it would be better to write the title of the x-axis as ‘Total length (cm)’. In the caption to this figure (and Fig. 6), it would be preferable to write ‘size composition’ rather than ‘size class distribution’. In Fig. 4, the values have been scaled both to a minimum of zero and a maximum of one, rather than just a maximum of one. Such scaling hides the true range of the monthly proportions of otoliths with clear margins, and for this reason is inappropriate. Confidence intervals of the proportions should be displayed on the figure, together with details of the number of otoliths examined within each month (e.g., using a number adjacent to the point estimate of each monthly proportion). The caption to Fig. 5 could be written as ‘Lengths at integer ages for Mycteroperca olfax in the Galapagos with fitted von Bertalanffy growth curve’. The y-axis for Fig. 5 should be extended to include the origin. The border around the legend is distracting; I recommend that this border is removed. In the legend for Fig. 6, write ‘f’ and ‘m’ as ‘female’ and ‘male’, respectively. In the caption for Fig. 7, write the scientific name in italics. In Fig.7a, the arrow appears to point to connective tissue within one of the lamellae, rather than the lamella itself. In Fig. 7d, it might be useful also to indicate the presence of other cell structures, e.g., spermatids.
Relevant data were available for examination while undertaking the review.

Experimental design

The paper produces data that fill an identified gap in knowledge. It is likely to be of value given the status of the species and its listing as Vulnerable by the IUCN.
More detail of the sampling protocol should be provided. How were trips and ports selected for sampling and how were samples of catches selected to ensure that the sampled fish were representative of the total catch? What fishing gear selectivity, or distribution of fishing, might have affected the size composition such that the catch is not representative of the population? Do fishers retain all M. olfax that are caught, or are some discarded and, if so, what bias might this introduce?
What developmental stages of the gonads were considered mature? Provide evidence that it was possible to distinguish regenerating ovaries from those of immature fish, and thereby, when considering the relationship with length or age of the proportions of fish with mature gonads, justify the use of data for all sampled fish rather than just those of fish caught during the spawning period.
As with marginal increment analysis, edge analysis should be undertaken separately for fish with one band, two bands, etc. (see Campana, 2001, J. Fish. Biol. 59: 197–242). As noted earlier, the scaling of the monthly proportions between zero and one is inappropriate as it hides the true range over which those proportions vary throughout the year.
Details of the error in counts of growth bands for otoliths with different numbers of growth bands should be presented, together with measures of the consistency among readings, e.g. CV or APE (Campana et al., 1995, Trans. Am. Fish. Soc., 124: 131–138). Were different readers involved?
The authors provide no data relating to the age at which the first growth zone is formed, and thus whether the ages that are assigned are correct.
Explain how the power function was fitted to the weight and length data. Were the data logarithmically transformed, or were the data fitted assuming a log-normal distribution? Has a correction been made for the bias associated with back-transformation? Is the coefficient of determination, which is reported in the results, that of the log-transformed data?
It would have been appropriate to explore how the gonadosomatic index (standardised for length) varied among the different months, and to have considered the duration of the spawning season when determining the relationship between proportion mature and both length and age, but particularly the latter variable. The logistic model that is used to describe the relationship between proportion mature and total length is incorrect as it requires both a location and a shape parameter. That is, it should be ‘log[p/(1-p)] = a + b TL’. Explain how the model was fitted, e.g. which function in R was used.

Validity of the findings

Comparison is made of the mean size of the overall sample with the mean size of the otolith subsample (paragraph starting at line 188) and of a subsample of 200 of 297 fish used when taking tissue samples (paragraph starting at line 201). Such comparison is invalid, as the samples are not independent. It would have been appropriate to compare the mean size of the otolith subsample with that of the fish excluded from that subsample, and similarly to compare the subsample used when taking tissue samples with that of the 97 fish excluded from that subsample.
The paucity of data relating to the males is such that logistical equations to estimate the length or age at sex change are unreliable. Fig. 9 should be deleted. Caution should be used when presenting estimates of the length and/or age at sex change such that the imprecision of these values is appropriately recognised. In particular, it is inappropriate to present these values to 1 decimal place and it would be preferable to include the symbol ‘~’ before the estimates of length and age at which females are likely to become males.
It is inappropriate to conclude that the population of M. olfax at the Galapagos has experienced severe overfishing on the basis of the low number of larger individuals or highly biased sex ratio. The latter could be a result of an haremic life style (noting that, as advised at line 290, such a sex ratio had been reported previously). Inferences based on large size confound the processes of growth and mortality. I would suggest that the authors consider estimation of total mortality using the Chapman-Robson (1960; Biometrics, 16: 354–368) method, applying this to the age composition of their sample, and then compare this estimate with an estimate of natural mortality derived from a model such as that of Hoenig (1983) or of Then et al. (http://icesjms.oxfordjournals.org/content/early/2014/08/19/icesjms.fsu136.full.pdf+html). The conclusion relating to overfishing is likely to be the same, but the basis for that conclusion will be more sound. The points made by the authors in lines 279–283 are pertinent, however, although the authors correctly identify other factors that might be involved.
The statement that edge analysis has demonstrated that the bands are formed annually is likely to be true only for younger fish due to the use of pooled data for all ages in the edge analysis and the low numbers of older fish within the samples.
Comparisons of point estimates of the Brody growth coefficient and asymptotic length of von Bertalanffy growth curves, such as appear in the paragraph starting at line 255, are inappropriate without considering the uncertainty associated with these and the correlation between the estimates of those two parameters, i.e. a high k is likely to be associated with a low asymptotic length. It is often more meaningful to compare the estimated lengths at selected ages, or present a figure showing the different growth curves.
The sentence at lines 256–257 presents a circular argument, as it assumes that M. olfax is long-lived then argues that its longevity of 24 years is consistent with other long-lived species.

Additional comments

Line 41. Replace ‘obtain’ with ‘attain’.
Line 65. The term “1920’s” should be written as “1920s”, as it is not the genitive case and not an abbreviation. The correction should also be made elsewhere in the text.
Line 71. Insert a comma after ‘declines’.
Line 124. ‘Weighted’ should be written as ‘weighed’, both here and elsewhere in the text.
Line 126. Was each otolith sectioned through its primordium?
Line 150. Replace ‘outlined by (Newman & Dunk, 2003)’ by ‘outlined by Newman & Dunk (2003)
Line 159. Advise what error distribution was assumed when fitting the growth curve. That is, what was the form of the objective function?
Line 161.’Where’ should commence with a lower case ‘w’, and ‘t’ is the age of the fish, not ‘time’.
Line 162. Delete ‘rate’.
Line 169. Replace ‘constants determined empirically’ with ‘parameters of the model’.
Line 179. Specify whether the logarithms used in this paper (here and at line 174) are natural logarithms or logs to base 10.
Line 234. Replace ‘be closer’ by ‘is closer’.
Line 255. Delete ‘rate’.
Lines 263–266. The two ‘sentences’ at these lines need to be re-written.
Line 284. Rewrite ‘pelagics’ as ‘pelagic species’.
Lines 293–299. It would be appropriate to consider whether the sample that was examined in this study is representative of the entire population, and whether, if there has been an effect of exploitation by the artisanal fishers, this is only a local rather than population-wide effect.
Line 317. ‘Are desperately needed to ensure’ is too emotive, as the paper has not demonstrated this. I would suggest that these words are replaced by ‘would ensure’.
Line 351 and elsewhere in references. Ensure that scientific names are written in italics.
Line 379 and elsewhere in references. Check the use of capitals for each word.

Reviewer 3 ·

Basic reporting

See comments below to improve text

Line 30, insert the, before IUCN
Line 33, deletes its, replace with the
Line 40 – concern regarding large size of t0 and low value of k
Line 43 – change to – a mean age of 7.8 years
Line 49 – delete urge, replace with provide the impetus
Line 49 – replace long-due, with long-overdue
Line 74 – insert a before licensing
Line 82 – replace its, with this
Line 85 – insert the after to
Line 99 – delete of, replace with arrangements for
Line 114 – add to previous paragraph
Line 122 – insert each after from
Line 122-123 – why are otoliths washed in ethanol?
Line 124 – deleted weighted, replace with weighed
Line 133 – deleted weighted, replace with weighed
Line 142 – check possible error in referencing format
Line 161 – Lt is the predicted mean length, amend
Line 161 – is the asymptotic mean length (not maximum), amend
Line 163-164 – why use a Ford-Walford plot – why not fit the length-at-age data by minimising the sum of squares to estimate values of Linf, k and t0
Line 167-169 – when the length-weight data were fitted, was there a correction undertaken to account for the bias in log transformation?
Line 203 – replace subsample with subsampled
Line 225 – delete should
Line 234 – insert may before be
Line 240 – insert to interpret after easy
Line 243 – add years to references
Line 249 – delete sustainability – replace with management arrangements
Line 259 – insert a, after in
Line 274 – delete Froese’s equations – replace with the equation of Froese
Line 285 – delete particularly be, replace with reflect

Experimental design

I have some queries in regard to the analyses.
Why use a Ford-Walford plot – why not fit the length-at-age data by minimising the sum of squares to estimate values of Linf, k and t0
The ages of M. olfax need to be amended to a decimal age through the use of an assigned birth date. Then an adjustment can be made to each whole age depending on the time of sampling. For example, if we assume a birth date for all fish of 1 September (i.e. the start of the spawning season) then all subsequent ages can be adjusted to when the fish was sampled. That is, if a fish of 15+ years was caught on 1 December, its decimal age would be 15.249 yrs.
If the length-at-age data is smoothed using decimal ages, then the estimate of t0 is usually more robust and thus the estimate of k is more accurate. Note that small sample sizes of young fish can also influence these parameters. Thus caution is needed in the use of these data and their application noting the above.

Validity of the findings

As noted above in the experimental design section - If the length-at-age data is smoothed using decimal ages, then the estimate of t0 is usually more robust and thus the estimate of k is more accurate. Note that small sample sizes of young fish can also influence these parameters. Thus caution is needed in the use of these data and their application.
Also need to note that the overall sample size of 141 fish is low and that additional sampling may generate more robust estimates in the future.

Additional comments

The work presented in this manuscript represents a new contribution to the biology of the Galapagos sailfin grouper. The contribution is significant and of broad interest. The manuscript is generally clear and concise and is presented in a logical sequence with sound conclusions. The article should be accepted after addressing the concerns raised above.

---

## Round 0.2 · accepted · Accept

Thank you for giving us the opportunity to publish your work.